Polymorphism in a Neotropical toad species: ontogenetic, populational and geographic approaches to chromatic variation in Proceratophrys cristiceps (Müller, 1883) (Amphibia, Anura, Odontophrynidae)

http://orcid.org/0000-0001-7664-6236 Vieira Kleber Silva 1 2 3 ksvieira04@yahoo.com.br
Santos Oliveira Erivanna Karlene 1
Vieira Washington Luiz Silva 1
http://orcid.org/0000-0001-6824-0797 Alves Rômulo Romeu Nóbrega 1 2
1 Laboratório de Ecofisiologia Animal do Departamento de Sistemática e Ecologia, Universidade Federal da Paraíba , João Pessoa, Paraíba , Brazil
2 Departamento de Biologia, Universidade Estadual da Paraíba , Campina Grande, Paraíba , Brazil
3 Programa de Desenvolvimento Científico e Tecnológico Regional (FAPESQPB), Universidade Estadual da Paraíba , Campina Grande, Paraíba , Brazil
Hedrick Ann
Electronic publication date: 2022 Mar 25
Publication date: 2022
Volume: 10
Electronic Location ID: e12879
Received 2021 Oct 26; Accepted 2022 Jan 12
Copyright: © 2022 Vieira et al.
Copyright year: 2022
Copyright holder: Vieira et al.
License: This is an open access article distributed under the terms of the Creative Commons Attribution License, which permits unrestricted use, distribution, reproduction and adaptation in any medium and for any purpose provided that it is properly attributed. For attribution, the original author(s), title, publication source (PeerJ) and either DOI or URL of the article must be cited.
License URL: https://creativecommons.org/licenses/by/4.0/

Keywords: Amphibia, Chromatism, Polymorphism, Populations, Variation

Funding: Paraíba State Research Foundation (FAPESQPB) FAPESQ/MCT/CNPq 313232/2015-6 and 300324/2018-9 This study was financed by the Paraíba State Research Foundation (FAPESQPB). Edital 002/13—FAPESQ/MCT/CNPq. Proc.: 313232/2015-6 and 300324/2018-9 for the post-doctoral scholarship. The funders had no role in study design, data collection and analysis, decision to publish, or preparation of the manuscript.

==============================
Quantifying variability is important for understanding how evolution operates in polymorphic species such as those of the genus Proceratophrys Miranda-Ribeiro, 1920, which is widely distributed in South America. P. cristiceps distribution is limited to the Caatinga biome in Brazil. We examined its chromatic variation from a populational perspective, looking at different phenetic polymorphism levels and probable chromotypic association by applying statistical and GIS tools that could facilitate future taxonomic research regarding this and other species. We characterized P. cristiceps colour patterns and re-evaluated its geographic variation, highlighting potential consequences for the taxonomy of the genus. Our results revealed six principle chromotypes whose frequencies varied among sex and ontogenetic classes. Phenotypic expression appeared to respect defined proportions and evidenced selective value for the species. We conclude that individual variation, together with typological traditionalism may overestimate the polymorphic magnitude at the population level and cause taxonomic inflation. Our data support the usefulness of P. cristiceps as a model for microevolutionary studies.

Introduction

Morphological variation plays a fundamental role in the evolution of species. Although not all characteristics are heritable, natural selection can potentially act on those that are transmissible to new generations (Ridley, 2004). Understanding how evolutionary mechanisms operate on populations through individual morphological variability has been the main objective of large number of different studies ever since the times of Darwin (Futuyma, 1987; Huxley, 1940).

Such studies seek, in principle, to understand the origin of biodiversity, and how it can be accessed from the recognizable and comparable differences and similarities among organisms. Assessing morphological variation in an operationally adequate approach making use of different techniques, methods, or philosophies has proven to be a huge challenge by taxonomic, or even conservationist, criteria (Coyne, Orr & Futuyma, 1988; Isaac, Mallet & Mace, 2004; Padial et al., 2010; Sokal, 1973; Zachos, 2016), principally in polymorphic species such as those of the genus Proceratophrys.

The genus Proceratophrys Miranda-Ribeiro, 1920, comprises a group of South American amphibians popularly known as small ox-toads, or horned minor frogs. They are widely distributed across Brazil and also occur in Argentina and Paraguay (Frost, 2021; Napoli et al., 2011). The taxon was traditionally difficult to classify, as its species have been consistently confused with those of the genus Ceratophrys Wied-Neuwied, 1824, and often placed within the same genus (Boulenger, 1882; Braun, 1973; Gravenhorst, 1829; Günther, 1873; Miranda-Ribeiro, 1920, 1923; Müller, 1884).

The genus Proceratophrys was originally described by Miranda-Ribeiro (1920) based on the presence of a “dilated post-tympanic bone, spiculated eyelid and the absence of a keratoid appendix” as diagnostic characteristics. The species included in the description were P. appendiculata (Günther, 1873); P. boiei (Wied-Neuwied, 1824); P. cristiceps (Müller, 1883) and P. renalis (Miranda-Ribeiro, 1920). Miranda-Ribeiro highlighted a number of morphological traits, some of which were morphometric and others chromatic.

The genus has been revised several times due to taxonomic ambiguities, and the validity of some species has been questioned (Barrio & Barrio, 1993; Dias et al., 2013; Kwet & Faivovich, 2001; Lynch, 1971). A total of 42 Proceratophrys species are currently recognized (Frost, 2021), distributed in different biomes and morphoclimactic dominions (such as the Amazon, Caatinga, Cerrado, Chaco, Atlantic Forest and Pampas) (Barrio & Barrio, 1993; Giaretta, Bernarde & Kokubum, 2000; Izecksohn et al., 1998; Martins & Giaretta, 2011). Although Proceratophrys cristiceps has been considerably discussed during the last decade, its proposed taxonomy still arouses doubts (Cruz, Nunes & Juncá, 2012; Martins & Giaretta, 2013), and the distribution suggested by Junior et al. (2012) and Mângia et al. (2020) contains somewhat dubious and questionable records—largely reflecting that some are syntopic with other species in the goyana group, or that they were encountered in unusual biome for that species.

Similar to some other anurans, P. cristiceps displays chromatic and morphometric polymorphism (Vieira & Vieira, 2012). At least two chromotypes have been described for this species (Vieira et al., 2008), which may reflect environmental fluctuations and/or genetic events on populations (Dias & Gonçalves da Cruz, 1993; Smith & Skúlason, 1996). This information has largely gone unnoticed in recent studies, but if extended to other species, it may partly explain the taxonomic inflation (Aleixo, 2009; Alroy, 2003; Isaac, Mallet & Mace, 2004; Padial & De la Riva, 2006) observed in the genus in recent decades, with high numbers of species being described in a short period of time in the absence of accurate taxonomic revisions (Junior et al., 2012).

Chromatic variability is common in anurans (Hoffman & Blouin, 2000; Kakazu, Toledo & Haddad, 2010), and facilitates their survival in areas with many predators (Bourke, Busse & Bakker, 2011). In such cases, chromatic polymorphism may provide a wide range of variation that enables, for example, avoidance of visual recognition by creating patterns that tend to match natural substrates in their environments (Duellman & Trueb, 1994; Hoffman & Blouin, 2000). At the taxonomic level, however, chromatic variation can generate confusion among taxonomists, although precise descriptions of external characteristics (such as intra- and interspecific colouration patterns), could potentially reduce or even resolve serious species identification or classification problems (Grismer, Wong & Galina-Tessaro, 2002).

The importance of polymorphism in certain organisms, which may include P. cristiceps (Arruda, Costa & Recco-Pimentel, 2017; Arruda et al., 2012), can reside in improving the adaptative capacities of individuals in relation to environment stress and/or predation (Ridley, 2004). Natural populations are constantly exposed to widely variable conditions, and regardless of the degree of accumulated or displayed differences among them, one limiting factor for individual survival will be morphophysiological adaptability (Ricklefs, 2008). The survival and adaptation of populations of organisms will depend on the maintenance of viable numbers of individuals on which evolutionary mechanisms can act and give rise to what we call biodiversity (Badii et al., 2007; O’Neill, Beard & Pfrender, 2012).

In light of the importance of populational polymorphism in taxonomic and evolutionary research, we have sought to precisely examine the chromatic variation in our model organism, P. cristiceps, and to determine the extent of its chromatic variability at both ontogenetic and population levels by searching for explanatory patterns (including morphometric) along defined geographic gradients that could corroborate or bring into question certain taxonomic proposals. We also attempt to heuristically explain the origin of the variability found, and produce information that could facilitate identifying the species and their congeners, and thus favour future studies of the ecology, biogeography and systematics of the genus—as well as of other species.

Materials and Methods

Origin of the examined material

A total of 634 Proceratophrys cristiceps specimens from 37 localities were analyzed (Appendix). All the individuals were available in the Animal Ecophysiology Laboratory (UFPB) and the Herpetological Collection of the Universidade Federal da Paraíba (CHUFPB). The taxonomic identities of the samples were verified by consulting descriptions and diagnoses (Cruz, Nunes & Juncá, 2012; Müller, 1884). It was possible to identify excellent samples from different areas in northeastern Brazil in those collections, and their geographic information proved to be important for producing habitat suitability and spatial similarity maps for the study species.

Sexual and ontogenetic identifications, and stages of specimen development

The sexes of the preserved animals were identified by making an incision in the posterior ventrolateral portion of the abdomen with the aim of inspecting their sexual structures: ovaries or testicles (Heyer, 2005). The ontogenetic classification adopted herein for the metamorphosed animals follows Izecksohn & Peixoto (1980, 1981) and Mercadal de Barrio & Barrio (1993). The individuals considered as juveniles were those with cloacal-rostrum lengths ≤25 mm; subadult lengths were from 26 to 35 mm, whereas adults had lengths ≥36 mm. The compatibility of those classes with the maturity of the individuals (animals potentially capable of competing for sexual partners) was tested using Pearson’s Chi-square test of independence or association. We constructed two frequency tables, organized so that each cell represented a unique combination of specific values (cross tabulated), which allowed us to examine the frequencies of observation belonging to the determined categories in terms of more than one variable. Examinations of those frequencies allowed the identification of relationships (statistically significant or not) among the categories. The maturity of individuals was determined by their gonadosomatic index (the correlation between length and gonadal volume), oocyte type and the presence of developed and voluminous fatty bodies (Costa et al., 1998; Duellman & Trueb, 1994; Noble, 1931; Tolosa et al., 2014).

The aforementioned classification enabled us to identify operational ontogenetic units (OOUs) consistent with each suggested post-larval developmental phase (Vieira & Vieira, 2012), whose chromatic patterns were statistically consistent with respect to the analyzed frequencies (without distortions caused by small samples). In the case of the local populations studied (sensu Mayr, 1977), the ontogenetic categorization used herein expanded our understanding of variation in P. cristiceps, at both chromatic and morphometric levels.

Chromatic characterization

The chromatic characterization of both living and preserved Proceratophrys cristiceps specimens was performed based on the standardisation suggested by the colour catalogue for field herpetologists (Köhler, 2012) to decrease or avoid ambiguity issues in relation to the terminology and the description of the observed hues. The study of live animal colours was performed through in loco observations. The specimens preserved in alcohol (70° GL) were immersed in water to enhance the contrasts of their spots, stripes, and colouration under both natural and artificial light. That technique improved pattern identification as well as the descriptions and classifications of possible chromotypes.

The colours and dorsal spot patterns of P. cristiceps were recorded as digital images (DSC-H10 Sony, 8.1 Megapixels). All image captures were made at the same distance (25 cm) from the specimens with the camera lens in a horizontal position (using flash and a white background to highlight contrasts). We considered the numbers and sizes of the dark spots on the dorsal surface of the body of each specimen (Rabbani, Zacharczenko & Green, 2015). The dark spots were defined according to their contrast with the surrounding dorsal colour (Fig. S1B). The chromatic areas of the spots were calculated using ImageJ vol. 1.8.0 (Rasband, 2018). The images (.tiff) were processed, converted to 8-bit (grey value) files, and then quantified. The measurement interval was 0.1-infinity, which enabled calculating (in pixels/mm2) even the smallest spots (by gradient), considering the total body area of each specimen (Fig. S1C). The reference scale used was 20 mm.

Analysis of interpopulation chromatic and morphometric variation

Morphological variation, as a continuous or discrete property, can generate mistakes when certain categories and explanatory variables are disregarded in comparative tests. It is therefore necessary to first verify the magnitude of any likely variation in recognized variables and/or factors to avoid fragile comparisons and mistaken conclusions regarding their simultaneous effects (Zar, 2014).

Morphometric variation, for example, originates either from sexual dimorphism or ontogeny but is often not considered when comparable categories are separately (or simultaneously) tested in recognized populations. With that in mind, we attempted to identify different forms of variability in our samples and test them within and among the chromatic observed categories.

The morphometric (Fig. 1; Vieira & Vieira, 2012) and chromatic variations observed in Proceratophrys cristiceps were tested using multiway ANOVA with unequal replications and the Kruskal–Wallis test, the latter being indicated for samples with unknown distributions. Comparisons among frequency proportions were achieved through cross-tabulation and were carried out using Pearson’s Chi-square tests. That representation was found to be very informative, enabling us to re-examine the data in a simplified manner (line plots).

Figure 1 Measurements taken for the Proceratophrys cristiceps specimens (digital caliper/0.01 mm precision):

CRL, Cloacal Rostrum Length; ED, Eye Diameter; FoL, Foot Length; FL, Forearm Length; HaL, Hand Length; HL, Head Length; HW, Head Width; IMCL, Internal Metatarsal Callus Length; ID, Internarinal Distance; InD, Interocular Distance; NED, Nostril Eye Distance; ThL, Thigh Length; TL, Tibial Length; RND, Rostrum Nostril Distance. More details in Vieira & Vieira (2012) and Watters et al. (2016). Image credit: Kleber Vieira.

Population analysis

The collection localities were accepted here as true populations for strictly operational reasons. This was done with the intention of producing sub-samples, presumably considered as distinct populations (following the traditional definition that they need to be contiguous, but situated in different territories) separated by geographical gaps of relative lengths (Dobzhansky, 1970; Mayr, 1977). The premise then was that the separation of samples by location would generate exclusive and independent populational sets (with no interbreeding or gene flow between them).

We therefore decided to identify presumed breeding cross sets to mitigate methodological eventualities, or the “demes” (Gilmour & Gregor, 1939; Winsor, 2000). According to our terminological redefinition (with strictly operational application), a deme would be any cluster of local populations closely related because of sharing at least one exclusive characteristic (phenon), without necessarily supporting any possible taxonomic distinction at the species level, but conferring a particular identity (as it is more frequent and statistically significant).

We subsequently excluded localities with only one collected specimen (n = 6) to access part of the variability of the presumed populations (the phenons) through certain attributes (see below). We established 15 individuals per location as the minimum sample size due to circumstantial and operational limitations. We considered here a statement of the central limit theorem (Fischer, 2011) where, if χ has well defined mean values and deviations, the mean terms will present an approximately normal distribution, even though the samples are not large. We also applied a distribution method with the sample replacement of random means for two elements in situations where the samples presented values less than those established (Callegari-Jacques, 2004; Zar, 2014). Thus, the possible averages of the individual samples were randomly obtained (two by two) and replaced in order to compose probable samples, until the established operational limit was reached. Finally, the distribution was ordered and the relative frequency of each element calculated, as well as its position in Z.

The graphical representation of the distances between the demes had a multiple comparison matrix of Z values derived from the Kruskal–Wallis test as support. Next, we applied three-dimensional ordination of the coordinates in Cartesian space (Multidimensional Scaling metric). The choice of the number of dimensions was determined by the traditional scree test (Cattell, 1966), establishing seven dimensions at the stress levels obtained to adapt the quadratic matrix in the representation space. Our intention was to identify geographical signals in the clusters (Euclidean distance) along the dimensional axes to later compare them to the diversity mapping of the phenetic traits of the sample populations, which were conducted at the regional level and arranged in a 0.78° raster cell (86.56 km × 86.58 km along the line of the equator). The geographical similarity was calculated to compose a map based on the coefficients of variation of eight phenotypic traits (Hijmans, Guarino & Mathur, 2012; Scheldeman & van Zonneveld, 2010): chromatic (spot size; area occupied by the spot) and morphometric (CRL; HW; HL; FoL.; TL and ThL. See Fig. 1).

Principal components analysis was an option regarding population variation in our model species and was used to determine a factor that could simply explain the probable variability found, based on the possible linear combination of our variables.

We confirmed the normality of the residuals (probability-probability plot) and the symmetry of the multivariate population distributions prior to the analyses (Figs. S2 and S3). For the latter, we calculated Mardia’s multivariate skewness and kurtosis with tests based on Chi-square (skewness) and normal (kurtosis) distributions. All the tests were processed using the R v.3.5.0 basic package (R Development Core Team, 2018) and Past v.3.1.5 (Hammer, 2016) software.

In addition to the metric data, and to interpret probable variation among the demes, we collected information on several explanatory variables such as vegetation cover; climate, following the Köppen–Geiger classification (Peel, Finlayson & McMahon, 2016); altitude; rainfall; and temperature (min and max) of all of the locations where the specimens were collected. That information was obtained from the National Meteorological Institute (INMET, 2020) and from Freemeteo (2019). Both provide regular climatological data (monthly and annual means) in an historical series from 1960 until the present, with a minimum radius of 2 km distance for each UTM (Universal Transverse Mercator) coordinate.

Environmental niche modelling

The potential distribution maps were generated with the intention of interpreting the distribution of P. cristiceps in terms of determined and defined predictor variables. We used two software programmes with the goal of mitigating any possible effects on the distributions of a species with restricted vagility caused by heuristic factors, such as variation in growth rates, the principle of exclusion or coexistence probabilities with predators, and dispersion limits—see the BAM scenario (Soberón, 2007; Soberón & Nakamura, 2009): the DIVA-GIS (Hijmans et al., 2005) and the MaxEnt (Philips et al., 2017; Phillips, Dudik & Schapire, 2006). We then estimated the proportional quantity of probable presence based on the real records of the sample through MaxEnt (Soberón & Nakamura, 2009), balancing the effects caused by the models generated in Maxent in terms of sensibility vs. specificity (Jiménez-Valverde, 2012) with the BIOCLIM (DIVA-GIS). This was because BIOCLIM is capable of correctly estimating the probabilities of A (regions where the fundamental or potential niches areas occurs) and G0 (distribution area of the species where abiotic and biotic conditions are favourable and within reach of dispersing individuals) by including them in a relatively larger prediction compared to Maxent (Qiao, Soberón & Peterson, 2015).

Our predictions were generated through the information available in the WorldClim portal (Version 2.1), which were scenopoetic variables (temperatures and precipitation) with a range of annual means from 1970 to 2000 (Fick & Hijmans, 2017). All the maps presented herein are at a resolution of 30 arc seconds (~1 km2) in GCS WGS 1984 projections.

Checking the taxonomic functionality of phenetic characteristics

We analyzed the ambiguity and the frequencies of the diagnostic characteristics commonly used at the taxonomic level within the genus Proceratophrys. We tested the functionality of the information provided by the authors (see below) by comparing them to each other and with the phenotypic traits of our samples P. cristiceps individuals. We also checked the types of taxonomic features, and counted how many times they were applied by different authors (to different species). When one of those characteristics was recognized in our samples, or among the different authors, we could then verify the ambiguity of that phenetic trait. Our objective was to verify if identical diagnostic features could be found among distinct species (refutability principle). We constructed a matrix of meristic variables according to the frequency of the characteristics used. Next, we produced a set of common values from the available data based on six phenetic variables: colour; bone (considering the description of the head form); tissue (material: eye, eyelid, interdigital membrane, tympanum, tongue, vocal sac, warts, tubercles and nodules); measurements; sonogram and genetics (including karyotype).

We then generated a grouping in random blocks of partitioned density from the absolute values structured from k groups, so that the sets were brought together in a greater order of similarity (Hartigan, 1975). In this study we sought to identify significant patterns in the choice of specific features (by the authors) in descriptions and diagnoses that could define the underlying taxonomy. The studies consulted were Gravenhorst (1829), Günther (1873), Müller (1884), Miranda-Ribeiro (1937), Lynch (1971), Braun (1973), Jim & Caramaschi (1980), Izecksohn & Peixoto (1981), Barrio & Barrio (1993), Eterovick & Sazima (1998), Giaretta, Bernarde & Kokubum (2000), Gonçalves da Cruz, Prado & Izecksohn (2005), Ávila, Kawashita-Ribeiro & Morais (2011), Napoli et al. (2011), Martins & Giaretta (2011), Cruz, Nunes & Juncá (2012), Junior et al. (2012), Ávila, Pansonato & Strüssmann (2012), Brandão et al. (2013), Godinho et al. (2013), Martins & Giaretta (2013), Mângia et al. (2018) and Mângia et al. (2020). The sampling was performed in such a way as to unite all the information of the species in the controversial cristiceps group (Dias, de Carvalho-e-Silva & de Carvalho-e-Silva, 2014; Giaretta, Bernarde & Kokubum, 2000).

Results

Chromatic analysis

Our observations indicated the existence of at least six main chromatic variations in the Proceratophrys cristiceps (Fig. 2):

Figure 2 Chromatic variation in Proceratophrys cristiceps individuals.

The diversity found is characterised by the general colour pattern, saturation, and the distribution of dorsal patterns.

Chromotype 1 (n = 93, 15%): brown bichromatic colouration in diverse hues (C22–C25) on a tawny olive and drab brown background (C17 and C19), whose spots or stripes, sometimes distributed in a well-defined direction, impede the recognition of a characteristic dorsal geometric figure—“arrowhead” (Miranda-Ribeiro, 1937). Conspicuous suborbital bands. Animals moderately melanised, and with two or more interorbital stripes (often in contact, and with a lighter one in the middle). Generally occurring in leaf litter (98.48%);

Chromotype 2 (n = 271, 43%): similar to chromotype 1 in terms of having brown colouration and suborbital or interorbital bands (two bands, with one of being Y-shaped), however, there is a well-defined dorsal geometric figure laterally limited by dark bands (maroon—C38) in the orbit-cloaca direction. There are also lighter nuances on the flanks (salmon—C57 to C59) and on the limbs, stomach and snout (cyan white—C155). Usually occuring in leaf litter (97.02%) or gravel (2.98%);

Chromotype 3 (n = 39, 6%): with very clear brown-grey colouration, and slightly variegated (C256 to C259). Evident dorsal geometric figure and yellow-brown colouration (C84), distributed in the orbit-cloaca direction; limited by two bands (in opposing toothed arches) and lines of semi-parallel glandular nodules. Single interocular stripe and two well-defined suborbital stripes. May have discrete rusty tones (C253) in the supraocular portions and sides of the body. Generally occurring in earthy soil with sparse leaf litter (92.83%);

Chromotype 4 (n = 58, 9%): with evident trichromatic colouration, whose rusty red hue (C35 and C253) cover a large part of the body. Clear dorsal geometric figure with a pale-yellow colouration (C2 and C3), laterally limited by regular dark bands (C30) in an orbit-cloaca direction. Suborbital stripes are not clearly evident; presence of only one interocular stripe. A pineal spot present. There are also white hues (C155 and C261) in the lateral portions of the body and limbs, similar to Chrom2. Generally inhabiting sandy soils (6.25%), grit or gravel (93.75%);

Chromotype 5 (n = 51, 8%): general colouration monochromatic as compared to the others chromotypes, generally with rusty red hues (C57 and C58) or yellow-brown characteristic (C17). Barely visible spots or streaks. Generally occurs in grit or gravel (93.30%);

Chromotype 6 (n = 122, 19%): general brown-grey colouration (C19) with diverse nuances, with evident yellow-brown spots (or lighter hues C12 and C111) distributed in characteristic areas: snout and suprascapula. The dorsal geometric figure is laterally outlined by spots in a toothed arch shape, although not well defined. Generally inhabiting earthy or sandy soils (81.26%) and even in leaf litter (18.74%).

The frequencies of these chromotypes did not indicate dimorphic variation in the species, demonstrating an almost identical distribution between males and females, except for Chrom5, whose frequency in males was similar to Chrom4 (Fig. 3). Furthermore, we observed a proportional expression of the six phenotypes for each relative frequency of P. cristiceps (≈14:43:6:9:8:20), which was also maintained internally among the samples and localities (Table S1), suggesting that these phenotypes may be governed much more by heritable factors than by environmental or epigenetic ones (apparently by Mendelian inheritance).

Figure 3 Chromotypes of Proceratophrys cristiceps with a distribution of their frequencies varying in terms of sex, maturity and ontogenetic development.

The frequencies of Chrom5 were found to be higher in juveniles compared to sub-adults and adults when analyzing those same samples by ontogenetic class. We also verified the ontogenetic class frequencies for each sex—which demonstrated patterns with little difference from that of the species as a whole. Unlike females, the male chromotypic variations of Chrom3, Chrom4, and Chrom5 were significantly different, therefore moving away from the general species’ pattern (Fig. 4).

Figure 4 Chromotypes of Proceratophrys cristiceps with the distributions of their frequencies varying between sexes according to their maturity and ontogenetic development (post-larval).

The significant differences observed for the males suggest a curious and discreet effect of the factors acting on the sex variable.

The chromotypes also evidenced different frequencies in terms of maturity, with slightly lower frequency of Chrom4 and a higher frequency of Chrom5, mainly varying among mature individuals (Fig. 3). The variations revealed smaller numbers of adult Chrom4 individuals as compared to adult Chrom3 and Chrom5 individuals. Those differences were maintained in both males and females when analyzing the samples separately.

Another peculiarity of the studied specimens was their integumentary saturation (proportional quantity of dark in relation to light background). The Chrom5 individuals found studied here were less saturated than the others (Fig. 5), with a lower average size of the dorsal patches, and the area occupied by them (as well as their distribution) being reduced. Those variations, which characterised the form and extension of the dorsal designs, were significant and independent of sex, ontogenetic class and maturity, considering the species as a whole or internally among the samples (Figs. S4–S7).

Figure 5 Saturation of Proceratophrys cristiceps chromotypes.

The dorsal patterns are formed in accordance with the size of spots as well as their proximity to each other (distribution). The arrows represent derivation hypotheses, wherein Chrom2 is indicated as a basilar or heterozygous pattern (higher frequency, design complexity, and moderate saturation). Scatterplot graph for the mean saturation values (mm2) highlighted. Bar: 25 mm.

Although the distribution of dorsal spots did not vary significantly between males and females, the average size (in mm2) was greater in females; they were also more saturated than those of the males (Figs. S4–S7). The Chrom6 juvenile females (but not Chrom6 males) were very different from the other chromotypes, as their spots were observed to be larger.

Morphometric analysis and phenetic trait diversity

Males and females were morphometrically different in the general sample (except for ED, InD, FL and RND), but those variations were absent in juveniles and even in sub-adults (Figs. S8–S11). Males and females did not differ morphometrically in the permutations performed in terms of chromotypes. Only adult males (Chrom6 and Chrom3) or mature males (Chrom3, Chrom1 and Chrom2) differed from each other in the internal analysis of the samples, with differences being observed in the cephalic region (RND, ED, HL and HW) and in relation to the internal metatarsal callus.

When examining the coordinate factors based on correlations, only Chrom5 and Deme5 were more concentrated in the superior portion of the second component (Fig. S12); the others were almost uniformly distributed in the Cartesian space, without any variable (active or supplementary, morphometric or chromatic) supporting the composition of the demes, and they were not easily explained by the environmental predictors. Geographically supported and consistent groups were produced, however, when the multidimensional scaling diagram was associated with the phenetic trait diversity mapping. The results indicated Almas and São Mamede; Serra Talhada and Caicó; Junco and Jaguaribe; Cabaceiras and São João do Cariri as markers of zones with shared phenons (Fig. 6), constituting a strong indicator of the occurrence of genetic flow between populations.

Figure 6 Mapping of demes obtained by multidimensional scaling using Z value similarity of the relative Kruskal–Wallis scores.

Start config.: Guttman-Lingoes. Area occupied by dorsal spots (A) and Mean size of dorsal spots (B). 1. Almas; 2. Arcoverde; 3. Boa Vista; 4. Cabaceiras; 5. Caicó; 6. Caracol; 7. São João do Cariri; 8. Serra das Confusões; 9. São José dos Cordeiros; 10. Crato; 11. Desterro; 12. Exú; 13. Jaguaribe; 14. João Câmara; 15. Junco; 16. Nascente; 17. Paulo Afonso; 18. Patos; 19. Pedra da Boca; 20. Quixadá; 21. São Mamede; 22. Serra Talhada; 23. Trindade; 24. Ubajara; 25. Várzea da Conceição; 26. Buíque; 27. Macaíba; 28. Santana dos Matos; 29. Serra de São Bento; 30. Santa Quitéria.

The phenetic trait diversity mapping indicated the existence of at least five demes in the P. cristiceps species (Fig. 7B) that were exclusively distributed in the Caatinga biome and transition phytophysiognomies, according to ecological niche modelling. The species is most likely found in predominantly arboreal-shrubby vegetation, under direct influence of precipitation and annual minimum temperatures (Fig. 7; Fig. S13).

Figure 7 Distribution of Proceratophrys cristiceps within the Caatinga biome and in transition areas.

According to the results of environmental niche modelling (ENMs) (A) and the mapping of their demes (B) based on the geographic similarity of the covariance of eight phenotypic traits (chromatic and morphometric).

Discussion

The probable meaning of variation in P. cristiceps

Species are a multidimensional phenomenon (Harold, 2002; Zachos, 2016), and studies of variation in organisms can provide essential information for the field of experimental taxonomy (Sneath & Sokal, 1973; Sokal & Rohlf, 1995), and consequently for systematics, biogeography and ecology. Taxonomic characteristics (defining or diagnostic) must therefore be thoroughly discriminated and understood, especially with respect to probable intraspecific variation.

In order to be able to deal with this probable variation, categories must be defined that are equivalent in experimentally comparable ways, so that any possible effects of simultaneous interaction between factors in terms of specific variables can be identified. Thus, it is not difficult to perceive that variation can be expressed as altered phenotypes that, within morphological limits, can determine the different forms that we normally identify when dealing with individual and populations (Nicoglou, 2015). With this in mind, our results indicated two clear levels of variation in P. cristiceps: morphometric and chromatic, with both having apparent and substantial adaptive value.

The chromatic variation observed in Proceratophrys cristiceps presumably has selective value, as the differential frequencies of the chromotypes suggest certain advantages to individual survival. The distribution of animals on different soil types seems to play a predominant role in the observed bias of chromotypic frequencies throughout post-larval development, and may indicate some type of frequency-dependent selection (Bond, 2007). In this case, the numbers of less saturated animals decreases as maturity or adulthood is reached, suggesting that certain phenotypes may be reinforced by local edaphic conditions (Fig. 8; Fig. S14), and that crypsis may have an importante role (Bonte & Maelfait, 2004; Endler, 1981; Moreno-Rueda, 2020; Rabbani, Zacharczenko & Green, 2015). The contrasting colours and spots create disruptive patterns that could function, when combined with general colouration and saturation, as a highly effective strategy against predators (Cuthill et al., 2005). Together, the two mechanisms (disruptive colour and crypsis) may at least partially explain the observed variation in their frequencies, especially among juveniles, although they cannot explain the relative sample proportionality.

Figure 8 Juveniles of Proceratophrys cristiceps observed in the Pedra da Boca State Park.

(A) Chrom3; (B) Chrom6; (C) Chrom4 and (D) Chrom5. The contrasting colourations in relation to the soil types suggests reinforced adaptability in individual survival abilities (crypsis and disruptive colouration). Photo credit: Kleber S. Vieira.

As the chromatic expressivity (the observed percentage of a given phenotype) found in Proceratophrys cristiceps was not exclusive to specific samples, but was maintained even within and among categories (Table S1), phenotypic divergence due to local effects (polyphenism) can be easily discarded as an alternative explanation for the patterns identified. We therefore deduce that the observed chromatic polymorphism is grounded in a strong genetic basis (White & Kemp, 2016) and is reflected in the differential abundance and almost invariability of the poly- or dichromatic chromotypes identified (Mângia et al., 2020; Nunes et al., 2015; Vieira et al., 2008).

Another important factor in relation to the biogeographic aspect of our results was the existence of demes (understood herein as conglomerate populations) that were morphometrically smaller (on the average) in the north-western (hotter and drier) regions of the Caatinga. The most likely explanation for that observation would involve temperature-associated effects (Fig. 9). That explanation appears plausible when considering the determinants of potential distributions (Fig. S13), with the mean annual minimum temperature and the precipitation of the last quarter of the year significantly contributing to the habitat suitability model.

Figure 9 Morphometric gradients (cline and isophenes) observed in the distributions of the Proceratophrys cristiceps populations analysed.

The interpolation of the length values (cloacal rostrum distance) indicated that smaller individuals are found in the north-western region of the Caatinga (C), where temperatures are higher. Maps of South America showing the average annual temperature (A) and maximum temperature of the hottest month (B) for the years 1970–2000. The outlined area indicates the extent of the Caatinga biome. Climate data source: Fick & Hijmans (2020).

While this cline effect appears to point to Bergmann’s rule (Bergmann, 1848; Blackburn, Gaston & Loder, 1999; Salewski & Watt, 2017), there is no clear concordance with anurans, where phenotypic plasticity controlled by genes may be involved (Ashton, 2002; Berven, 1982a, 1982b) and would favour adaptative strategies to avoid thermoregulatory imbalances and hydric stress, with geographic selection gradients (Endler, 1977; Stebbins & Cohen, 1995), in turn, conferring a low metabolic energy cost to the animals (Bernardo, 1994).

At a more restricted level, the morphometric variation observed in Proceratophrys cristiceps is partly a consequence of sexual dimorphism, ontogenetic effects (Vieira & Vieira, 2012), and cranial morphological alterations in response to adaptations to available food resources (Atencia, Solano & Liria, 2020; Brito et al., 2012; Emerson, 1985). The observed metric variations are negligible when compared to the chromotypes, either between sexes or among the developmental categories (maturity and ontogenesis as described herein). Thus, although chromotypic variation is evident and quite informative in P. cristiceps, it could be deceptive and lead to serious taxonomic problems if misinterpreted and examined in isolation. Thus, the evolutionary implication of variation (whether chromatic or morphometric) is difficult to approach experimentally, and taxonomic studies often view operational morphological units (OMUs) as different sub-species or even species. There are also underlying factual (and experimental) requirements necessary to explain the morphological divergence and the alleged taxonomic diversity (Van Holstein & Foley, 2020), where taxonomic richness is clearly correlated with rates of intraspecific population divergence.

Taxonomic implications of variation in P. cristiceps

There have been significant increases in the numbers of species descriptions in the genus Proceratophrys over the past 20 years. The taxonomic inflation rate between 2011 and 2021 was 45% (Fig. S15A), with the cristiceps group (with reduced eyelid appendages), which inhabits open and dry environments in the Cerrado and Caatinga (Dias, de Carvalho-e-Silva & de Carvalho-e-Silva, 2014) reaching 12% (Fig. S15B). Although the taxonomy of Proceratophrys cristiceps (and other species of the genus) has been studied and debated for decades (Barrio & Barrio, 1993; Cruz, Nunes & Juncá, 2012; Lynch, 1971; Mângia et al., 2020), it is difficult to determine if this increase in group diversity reflects true species diversity or only a typification of the intraspecific variability already observed (Junior et al., 2012).

When revisiting the original species descriptions, it could be seen that not only body coloration, but also the amount, sizes and appearances of nodules and tubercles are among the most common diagnostic (or defining) characteristics for all species in the cristiceps group (and others groups as well) (Fig. 10)—suggesting that the species were defined based on traits evidencing significant phenotypic plasticity.

Figure 10 Authors grouped based on the identification of regions with high densities of similar values (Two-Way Joining).

Clusters generated through the diagnostic use of identical phenetic traits. The highlighted blocks in light colours reflect greater sets of tissue characteristics (mainly nodules, warts and tubercles) used in the descriptions of the species of the genus Proceratophrys. Threshold Computed: 5.46 (St. Dv./2). Number of Blocks: 44. Total Sample Mean: 9.65. Standard Deviation: 10.92. The score on the right is the number of groups by the number of k-observations. The data indicate that certain categories of phenetic traits have been used uncritically (reflecting taxonomic traditionalism), which has led to a dependence on variable features. (a) Gravenhorst (1829); (b) Miranda-Ribeiro (1937); (c) Lynch (1971); (d) Jim & Caramaschi (1980); (e) Eterovick & Sazima (1998); (f) Ávila, Kawashita-Ribeiro & Morais (2011); (g) Napoli et al. (2011); (h) Günther (1873); (i) Müller (1884); (j) Cruz, Nunes & Juncá (2012); (k) Mângia et al. (2020); (l) Braun (1973); (m) Izecksohn & Peixoto (1981); (n) Mângia et al. (2018); (o) Barrio & Barrio (1993); (p) Caramaschi (1996); (q) Giaretta, Bernarde & Kokubum (2000); (r) Junior et al. (2012); (s) Brandão et al. (2013); (t) Martins & Giaretta (2013); (u) Gonçalves da Cruz, Prado & Izecksohn (2005); (v) Godinho et al. (2013); (w) Martins & Giaretta (2011); (x) Ávila, Pansonato & Strüssmann (2012).

Our observations, for example, indicated that nodules (including warts and tubercles) are extremely variable in terms of numbers, shapes and distributions, either isolated or regionally, on the same individual or among specimens (Figs. S16 and S17). Some animals have large and round nodules; distributed regularly or irregularly; with glandular appearances and salient; or smaller and more conical, or even flat—but they serve little purpose as defining or diagnostic characteristics of the chromotypes. In addition to the nodules, the shape of the snout, when viewed laterally or dorsally, was equally variable, due not only to allometric factors (Vieira & Vieira, 2012), but also in terms of the position of the specimens in the viewing plane. The difficulties encountered while using this information has also been discussed by other taxonomists (Brandão et al., 2013).

Another common characteristic used in descriptions of these species are the rows of opposite oculum-dorsal nodules and their associated spots and stripes. Those rows appear to be important in forming the arrowhead shape of the dorsal design (Miranda-Ribeiro, 1937). This shape becomes much less distinct, however, when those rows are discontinuous and dissolve into patterns of irregular spots and bands (very variable among individuals) that interconnect at various points, especially in the middle dorsal portion (Chrom1). The nodules in those discontinuities can spread in the suprascapular direction and the flanks of the animal, forming sinuous (or bifurcated) designs, with the larger branch sometimes expanding to the sacral area. This is usually evident in Chrom5 individuals.

We assume that specialists have been constrained by a typological traditionalism (see Fig. 10) that seems to interfere with their perception and forces them to choose more traditionally used morphological traits, while ignoring their evident plasticity or ambiguity. The consequence of acting in that matter (i.e., disregarding probable variation) is that species descriptions may not be sustainable in reality (Dobzhansky, 1970; Mayr, 1996).

By reviewing the descriptions of the species of genus Proceratophrys and comparing the information provided by the authors with each other and with the characteristics of the individuals in our samples, and then testing the probable ambiguity of the proposed diagnostic traits, it became evident that some species described in recent decades are not actually morphologically different from P. cristiceps or P. goyana, or even among themselves—as, for example, P. carranca (Godinho et al., 2013), P. branti (Brandão et al., 2013), P. huntingtoni (Ávila, Pansonato & Strüssmann, 2012) and P. dibernardoi (Brandão et al., 2013). Similarly, the same chromatic varieties observed in P. cristiceps may be equally recognisable in their congeners (Ávila, Kawashita-Ribeiro & Morais, 2011; Brandão et al., 2013; Junior et al., 2012; Martins & Giaretta, 2013)—This leads us to the conclusion that those presumed diagnostic patterns are, to a greater or lesser extent, common to the genus as a whole.

The identification of species as being distinct in recent decades often presupposed the hypothesis of sympatric speciation in the absence of an evident vicariant element (Godinho et al., 2013; Mângia et al., 2018; Martins & Giaretta, 2013). This has been the case with taxa (cryptic) that share many similarities, but whose distinctions (mostly linked to colour, warts or tubercles, or sometimes by acoustic [not immune to variability] and genetic analysis) can be ambiguous and conceptually confusing. Additionally, those distinctions have not even been tested under any experimental model of diversification dynamics (Ajmal Ali et al., 2014; Annibale et al., 2020; Schindel & Miller, 2005; Van Holstein & Foley, 2020), where patterns of trait richness are equivalent to the rates of intraspecific population divergence (and would thus reinforce the divergence hypotheses). This is mainly the case for species of the P. goyana and P. cristiceps groups (Martins & Giaretta, 2011); but why not then for the P. biggibosa, P. boei and P. appendiculata groups, whose taxonomic histories depend on variable phenetic traits, while evidence of pre- or post-zygotic barriers or their biogeographies continue to be elusive?

We, therefore, suggest that future studies using traditional characteristics be based on preliminarily sampling and statistical testing to determine whether they are truly diagnostic. Likewise, we cannot discount the hypothesis of taxonomic inflation in the genus Proceratophrys, especially the cristiceps group, due to poorly interpreted population peculiarities emerging from microevolutionary processes (Amaro et al., 2012; Mângia et al., 2020) instead of a taxonomic quality, due to the simple and unfortunate confusion of methods and concepts.

Finally, we conclude that individual variation, together with typological traditionalism, may overestimate the polymorphic magnitude of variation at the population level and be the cause of taxonomic inflation in many anuran species. Our data also support the usefulness of P. cristiceps as a model for microevolutionary studies.

Appendix

Specimens examined

BAHIA| Paulo Afonso (–9.401130556°S.; –38.20623333°W): UFPB12112, UFPB12114, UFPB12115, UFPB12116, UFPB12118, UFPB12124. Santa Terezinha (–12.771725°S; –39.52416389°W): CHUFPB24169. CEARÁ| Crato (–7.229958333°S; –39.41229722°W): CHUFPB19690, CHUFPB20690. Ipu (–4.321944444°S; –40.71083333°W): UFPB6127. Jaguaribe (–5.901030556°S; –38.62215°W): CHUFPB19946, CHUFPB20656, CHUFPB20657, CHUFPB20675, CHUFPB20940, CHUFPB21058, CHUFPB22183, CHUFPB22188, CHUFPB22195, CHUFPB22233. Junco (–4.814325°S; –38.98613889°W): UFPB10033, UFPB10034, UFPB10035, UFPB10036, UFPB10037. Quixadá (–4.972555556°S; –39.01541389°W): CHUFPB19935, CHUFPB22177, CHUFPB22191. Santa Quitéria (–4.324272222°S; –40.14281111°W): UFPB10752, UFPB10759, UFPB10760. Ubajara (–8.616025°S; –37.16555833°W): CHUFPB19726, CHUFPB19729, CHUFPB19886, CHUFPB19925, CHUFPB19969, CHUFPB20654, CHUFPB20662, CHUFPB20671, CHUFPB20680, CHUFPB20681, CHUFPB20683, CHUFPB20792, CHUFPB20818, CHUFPB20820, CHUFPB20821, CHUFPB20822, CHUFPB20827, CHUFPB20830, CHUFPB20854, CHUFPB20876, CHUFPB20894, CHUFPB20896, CHUFPB20921, CHUFPB20930, CHUFPB20933, CHUFPB20938, CHUFPB20939, CHUFPB20943, CHUFPB20946, CHUFPB21056, CHUFPB21347, CHUFPB21349, CHUFPB21351, CHUFPB21355, CHUFPB22178, CHUFPB22179, CHUFPB22187, CHUFPB22190, CHUFPB22194, CHUFPB22201, CHUFPB22205, CHUFPB22217, CHUFPB22222, CHUFPB22225. PARAÍBA| Boa Vista (–7.260538889°S; –36.24889444°W): UFPB1571, UFPB1572, UFPB1573, UFPB1574, UFPB1575, UFPB1576, UFPB1577, UFPB1579, UFPB1580, UFPB1581. Cabaceiras (–7.469663889°S; –36.30575833°W): UFPB11266, UFPB11267, UFPB11268, UFPB11269, UFPB11270, UFPB11271, UFPB11272, UFPB11273, UFPB11274, UFPB11275, UFPB11276, UFPB6691, UFPB6692, UFPB6693, UFPB6694. Desterro (–6.875280556°S; –37.53213333°W): UFPB1582, UFPB1583, UFPB1584, UFPB1585, UFPB1586. Fazenda Almas (–7.470833333°S; –36.88083333°W): FA01, FA44, FA45, FA46, FA149, FA154, FA158, FA159, UFPB4267, UFPB4270, WLSV1308, WLSV1346, WLSV1349, WLSV1463, WLSV1470, WLSV1472, WLSV1474, WLSV1475, WLSV1476, WLSV1477, WLSV1485, WLSV1487, WLSV1488, WLSV1497, WLSV1505, WLSV1566, WLSV1567, WLSV1572, WLSV2021, WLSV2026, WLSV2042, WLSV2131, WLSV2170, WLSV2252, WLSV2259, WLSV2260, WLSV2339, WLSV2340, WLSV2341, WLSV2388, WLSV2391, WLSV2935, WLSV3007, WLSV3016, WLSV3017A, WLSV3018, WLSV3019, WLSV3031, WLSV3032, WLSV3303, WLSV3304, WLSV3305, WLSV3318, WLSV3319, WLSV3320, WLSV3321, WLSV3990, WLSV4057, WLSV4063, WLSV4091, WLSV4093, WLSV4095, WLSV4207, WLSV4208, WLSV4209, WLSV4237, WLSV4335, WLSV4365, WLSV4375, WLSV4388, WLSV4397, WLSV4398, WLSV4399, WLSV4411, WLSV4492, WLSV4493, WLSV4494, WLSV4515, WLSV4529, WLSV4530, WLSV4533, WLSV4604, WLSV4646, WLSV4647, WLSV4765, WLSV4766, WLSV4767, WLSV4768, WLSV4769, WLSV4770, WLSV4771, WLSV4772, WLSV4773, WLSV4774, WLSV4775, WLSV4776, WLSV4777, WLSV4778, WLSV4779, WLSV4780, WLSV4789, WLSV4791, WLSV813, WLSV814, Y039. Patos (–6.986025°S; –37.31695278°W): KSV041, KSV053, KSV055, KSV079, KSV113, KSV196, KSV232, KSV233, KSV237, KSV246, KSV247, KSV248, KSV251, KSV266, KSV278, KSV313, KSV319, KSV320, KSV321, KSV322, KSV325, KSV326, KSV327, KSV328, KSV330, KSV346. Pedra da Boca (–6.459583333°S; –35.67788333°W): KSV02, UFPB8423, UFPB8424, UFPB8425, UFPB8426, UFPB8427, UFPB8428, UFPB8429, UFPB8430, UFPB8431, UFPB8432, UFPB8433, UFPB8434, UFPB8435, UFPB8436, UFPB8437, UFPB8438, UFPB8439, UFPB8440, UFPB8441, UFPB8442, UFPB8443, UFPB8444, UFPB8445, UFPB8446, UFPB8447, UFPB8448, UFPB8449, UFPB8450, UFPB8451, UFPB8452, UFPB8453, UFPB8454, UFPB8455, UFPB8456, UFPB8457, UFPB8458, UFPB8459, UFPB8460, UFPB8461, UFPB8462, UFPB8463, UFPB8464, UFPB8465, UFPB8466, UFPB8467, UFPB8468, UFPB8470, UFPB8471, UFPB8472, UFPB8473, UFPB8474, UFPB8475, UFPB8476, UFPB8477, UFPB8478, UFPB8479, UFPB8480, UFPB8481, UFPB8482, UFPB8483, UFPB8484, UFPB8485, UFPB8486, UFPB8487, UFPB8488, UFPB8489, UFPB8490, UFPB8491, UFPB8492, YL005, YL013, YL101, YL117, YL135, YL144, YL173, YL238, YL280, YL283, YL293, YL325, YL348. São João do Cariri (–7.45825°S; –36.48094444°W): WLSV001, WLSV002, WLSV173, WLSV209, WLSV244, WLSV245, WLSV258, WLSV596, WLSV884, WLSV885, WLSV886, WLSV899, WLSV900, WLSV901, WLSV902, WLSV903, WLSV904, WLSV904, WLSV905, WLSV906, WLSV965, WLSV966, WLSV967. São José dos Cordeiros (–7.4675°S; –36.84327778°W): UFPB11253, UFPB11254, UFPB11255, UFPB11256, UFPB11257, UFPB11258, UFPB11259, UFPB11260, UFPB11261, UFPB11262, UFPB11263, UFPB11264, UFPB11265, UFPB5866. São Mamede (–6.893888889°S; –37.08300833°W): UFPB11686, UFPB11687. PERNAMBUCO| Arcoverde (–8.437488889°S; –37.04850556°W): UFPB9678, UFPB9679, UFPB9680, UFPB9681, UFPB9682, UFPB9683, UFPB9684, UFPB9685, UFPB9686, UFPB9687, UFPB9688, UFPB9689, UFPB9690, UFPB9691, UFPB9692, UFPB9693, UFPB9694, UFPB9695, UFPB9696, UFPB9697, UFPB9698, UFPB9699, UFPB9701. Bezerros: UFPB7098. Buíque (–8.616025°S; –37.16555833°W): CHUFPB19895, CHUFPB19903, CHUFPB19908, CHUFPB19920, CHUFPB19921, CHUFPB19977, CHUFPB19978, CHUFPB20672, CHUFPB20830, CHUFPB20833, CHUFPB20855, CHUFPB20868, CHUFPB20884, CHUFPB20924, CHUFPB21057, CHUFPB22174, CHUFPB22175. Exú (–7.511944444°S; –39.72388889°W): UFPB7208, UFPB7209, UFPB7210, UFPB7211, UFPB7212, UFPB7213, UFPB7214, UFPB7216, UFPB7217. Nascente (–7.883244444°S; –40.47074167°W): UFPB9670, UFPB9671. Serra Talhada (–8.014947222°S; –38.28990833°W): UFPB9655, UFPB9656, UFPB9657, UFPB9658, UFPB9659. Trindade (–7.741983333°S; –40.288475°W): UFPB9672, UFPB9673, UFPB9674, UFPB9676, UFPB9677, UFPB974. Várzea da Conceição (–6.472177778°S; –39.11150278°W): UFPB9661, UFPB9662, UFPB9666, UFPB9668, UFPB9664, UFPB9667, UFPB9665, UFPB9663. PIAUÍ| Cajueiro (–2.932194444°S; –41.34146389°W): UFPB7086. Caracol (–9.281308333°S; –43.32954722°W): GGS2-01, GGS2-02, GGS2-03, GGS2-04, GGS2-05, GGS2-06, GGS2-07. Paulistana (–8.115602778°S; –41.20048889°): UFPB9669. Piripiri (–4.354030556°S; –41.83985556°W): UFPB10339. Serra das Confusões (–9.223002778°S; –43.48978333°W): GGS560, GGS608, GGS656, GGS657, GGS658, GGS673, GGS674, CHUFPB19973, CHUFPB19986, CHUFPB20878, CHUFPB22176, CHUFPB22193, CHUFPB22215, CHUFPB22219, CHUFPB22221, CHUFPB22227. RIO GRANDE DO NORTE| Caicó (–6.454016667°S; –37.10038889°W): UFPB14903, UFPB14904, UFPB14905, UFPB14906. João Câmara (–5.449719444°S; –35.87196389°W): GGS01, GGS02, GGS03, GGS04, GGS05, GGS06, GGS07, GGS08, GGS09, GGS10, GGS11, GGS12, GGS13, GGS14, GGS15, GGS16, GGS17, GGS18, GGS19, GGS20, GGS21, GGS22, GGS23, GGS24, GGS25, GGS26, GGS27, GGS28, GGS29, GGS30, GGS31, GGS100, GGS101, GGS102, GGS103, GGS104, GGS105, GGS106, GGS107, GGS108, GGS109, GGS110, GGS111, GGS112, GGS113, GGS114, GGS115, GGS116, GGS117, GGS118, GGS119, GGS120, GGS121, GGS122, CHUFPB19900, CHUFPB19984, CHUFPB20872, CHUFPB21300, CHUFPB21844, CHUFPB21860, CHUFPB21884, CHUFPB22224, CHUFPB23174. Macaíba (–5.862877778°S; –35.35527222°W): CHUFPB19847, CHUFPB19948, CHUFPB19949, CHUFPB19953, CHUFPB19961, CHUFPB19966, CHUFPB19972, CHUFPB19974, CHUFPB19976, CHUFPB19980, CHUFPB19995, CHUFPB20679, CHUFPB20682, CHUFPB20684, CHUFPB20790, CHUFPB20802, CHUFPB20834, CHUFPB20842, CHUFPB20848, CHUFPB20858, CHUFPB20864, CHUFPB20866, CHUFPB20869, CHUFPB20874, CHUFPB20883, CHUFPB20900, CHUFPB20903, CHUFPB21063, CHUFPB21348. Santa Cruz (–6.189175°S; –36.09248333°W): CHUFPB21054. Santana dos Matos (–5.964152778°S; –36.65888056°W): CHUFPB19938, CHUFPB20660, CHUFPB20840, CHUFPB20857, CHUFPB20890, CHUFPB20897, CHUFPB20928. Serra de São Bento (–6.418805556°S; –35.704275°W): CHUFPB22200, CHUFPB22203. TOCANTINS| Aliança (–11.37906111°S; –48.92268333°W): UFPB1588.

Supplemental Information

Supplemental Information 1 (A) A preserved specimen of Proceratophrys cristiceps (WLSV1463) immersed in water enhance the contrast of its spots and stripes.

(B) Characteristic dorsal (8-bit) chromatic pattern. (C) Total area of spots (red colour) calculated along the dorsal surface of the specimen. Measurements sets: area; minimum and maximum grey value; mean grey value. Bar: 56 mm. Photo credit: Kleber Vieira.

Click here for additional data file.

Supplemental Information 2 Normality of the residues and relative morphometric symmetry in the multivariate population distributions (probability-probability plot).

Click here for additional data file.

Supplemental Information 3 Normality of residues and relative morphometric symmetry in the multivariate population distributions (probability-probability plot).

Click here for additional data file.

Supplemental Information 4 Average size of the dorsal spots of Proceratophrys cristiceps females in terms of maturity and ontogenetic class (post-larval).

Chrom5 individuals are significantly different (α = 0.05) from the other chromotypes, demonstrating smaller spots. Curiously, females generally demonstrated a greater average spot size compared to males.

Click here for additional data file.

Supplemental Information 5 Areas occupied by dorsal spots of Proceratophrys cristiceps females in terms of maturity and ontogenetic class (post-larval).

Chrom5 individuals are significantly different (α = 0.05) from the other chromotypes, demonstrating smaller spots that are located farther apart from one another.

Click here for additional data file.

Supplemental Information 6 Average sizes of the dorsal spots of Proceratophrys cristiceps males in terms of maturity and ontogenetic class (post-larval).

Chrom 5 individuals are significantly different (α = 0.05) from the other chromotypes, demonstrating smaller spots. Some values not observed.

Click here for additional data file.

Supplemental Information 7 Areas occupied by the dorsal spots of Proceratophrys cristiceps males in terms of maturity and ontogenetic class (post-larval).

Chrom5 individuals are significantly different (α = 0.05) from the other chromotypes, demonstrating smaller spots that are located farther apart from one another. Males exhibit a smaller average distribution area as compared to females. Some values not observed.

Click here for additional data file.

Supplemental Information 8 The multifactorial permutations of variance did not show significant morphometric differences (α=0.05) among the chromotypes of Proceratophrys cristiceps,.

indicating that males and females are equivalent when comparing them in terms of ontogenetic classes (post-larval). Wilks’ lambda = 0.81; F(117, 4400, 6) = 1.05; p = 0.34. Vertical bars denote 0.95 confidence intervals (weighted marginal means, some means not observed).

Click here for additional data file.

Supplemental Information 9 The multifactorial permutations of variance did not show significant morphometric differences (α=0.05) among the chromotypes of Proceratophrys cristiceps,.

indicating that males and females were equivalent when comparing ontogenetic classes (post-larval). Wilks’ lambda = 0.81; F(117, 4400, 6) = 1.05; p = 0.34. Vertical bars denote 0.95 confidence intervals (weighted marginal means, some means not observed).

Click here for additional data file.

Supplemental Information 10 The multifactorial permutations of variance did not show significant morphometric differences (α=0.05) among the chromotypes of Proceratophrys cristiceps,.

indicating that the males and females were equivalent when comparing maturity classes (Immature and Mature). Wilks’ lambda = 0.80; F(52, 1218, 2) = 1.33; p = 0.063. Vertical bars denote 0.95 confidence intervals (weighted marginal means, some means not observed).

Click here for additional data file.

Supplemental Information 11 The multifactorial permutations of variance did not show significant morphometric differences (α=0.05) among the chromotypes of Proceratophrys cristiceps,.

indicating that the males and females were equivalent when comparing maturity classes (Immature and Mature). Wilks’ lambda = 0.80; F(52, 1218, 2) = 1.33; p = 0.063. Vertical bars denote 0.95 confidence intervals (weighted marginal means, some means not observed).

Click here for additional data file.

Supplemental Information 12 Chromotypes (A) and demes (B) represented against the first two principal components scaled for morphometric and chromatic variables.

PC1 is correlated with size dimensions, whereas PC2 is correlated with saturation. It is possible to verify that Chrom5 and Dem5 are more concentrated and distributed along the superior portion of the second component, suggesting the presence of low saturated specimens. The environmental predictors did not explain the chromatic variance observed, indicating the existence of underlying operating factors.

Click here for additional data file.

Supplemental Information 13 AUC curves and Jackknife tests of the environmental variables of the climate model (default parameters) for Proceratophrys cristiceps.

The data indicated that the species is typical of the Caatinga, being found with greater probability in the tropical savanna and semi-arid climate zones of this biome, according to the Köppen–Geiger classification.

Click here for additional data file.

Supplemental Information 14 Proceratophrys cristiceps adults observed in the Patrimônio Nacional Fazenda Almas Private Reserve.

(A) Chrom1 and (B) Chrom2. The contrast of the animals’ coloring in relation to the soil suggests adaptive reinforcement of the individual survival capacity (crypsis). Photo credit: Washington L. S. Vieira.

Click here for additional data file.

Supplemental Information 15 Number of species described among three diverse genera of anuran amphibians (A) and among those of Proceratophrys (B).

The lines represent least squares regressions, while the numbers over the dots represent the periodic rate (%) of the descriptions (A). We found that the species of the genera Leptodactylus and Rhinella increased at similar rates over the decades, being later surpassed by Proceratophrys due to its faster rate of annual descriptions (A). When compared among congeneric groups (B), the highest description rates are observed in the cristiceps group. The bigibbosa group has been reasonably stable, but the boiei group rate has declined in relation to the total. Data obtained from Frost, D. R. (2021). Amphibian Species of the World: an Online Reference. Version 6.1.

Click here for additional data file.

Supplemental Information 16 Nodule variation in P. cristiceps (warts and/or tubercles) in terms of shape, type, and position.

Gular region: slightly globular and smooth (A) or rough (B); dorsal glandular nodules varying in shape and size (C and D); ventral posterior portion: elongated and flattened (E). Photo credit: Kleber Vieira.

Click here for additional data file.

Supplemental Information 17 Nodule variations in size, numbers, distributions, and positions (warts and/or tubercles).

on the outer portion of the right forearm and buccal (and/or subocular) commissure in specimens of P. cristiceps. A (WLSV 1474); B (WLSV 4095); C (WLSV 4791); D (UFPB 23174); E (UFPB 7214) e F (KSV 237). Photo credit: Kleber Vieira.

Click here for additional data file.

Supplemental Information 18 Proportions of chromotypic expression in Proceratophrys cristiceps.

The relative frequencies varied little among the sample categories analysed: ~14:43:6:9:8:20. Significant variations were not observed (α=0.05).

Click here for additional data file.

Supplemental Information 19 Raw data.

Click here for additional data file.

We are grateful for the helpful assistance of Fagner R. Delfim together with the Coleção Herpetológica do Departamento de Sistemática e Ecologia da UFPB and also to Professors Daniel O. Mesquita and Gustavo Vieira for their time, attention and access to specimens. To Professor Paulo G. Montenegro for always making the laboratory and materials available. We would also like to thank the reviewers and editors for their criticisms and recommendations.

Additional Information and Declarations

Competing Interests

Author Contributions

Data Availability

The authors declare that they have no competing interests.

Kleber Silva Vieira conceived and designed the experiments, performed the experiments, analyzed the data, prepared figures and/or tables, authored or reviewed drafts of the paper, and approved the final draft.

Erivanna Karlene Santos Oliveira performed the experiments, prepared figures and/or tables, authored or reviewed drafts of the paper, and approved the final draft.

Washington Luiz Silva Vieira performed the experiments, analyzed the data, authored or reviewed drafts of the paper, and approved the final draft.

Rômulo Romeu Nóbrega Alves analyzed the data, authored or reviewed drafts of the paper, and approved the final draft.

The following information was supplied regarding data availability:

The raw data and other data are available in the Supplemental Files.

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
