# Peer review of "Polymorphism in a Neotropical toad species: ontogenetic, populational and geographic approaches to chromatic variation in Proceratophrys cristiceps (Müller, 1883) (Amphibia, Anura, Odontophrynidae)"

_PeerJ, doi:10.7717/peerj.12879_

## Round 0.1 · original submission · Major Revisions

Please revise your manuscript to address all of the reviewers' concerns. In particular, please pay attention to the criticisms of your writing style. It might be a good idea to have a proficient English speaker review your writing.

Reviewer 1 ·

Basic reporting

Review: Proceratophrys biogeography and taxonomy

This paper focuses on the taxonomy and biogeography of the genus Proceratophrys in Brazil. On the positive side, the authors appear to have evaluated a variety of different morphological and chromatic characteristics for a large number of locations and populations, and they present some convincing arguments that previous researchers may have designated species without sufficient rigor or justification. On the negative side, this paper is very poorly written. The paper is riddled with what appear to be inappropriate conclusions, awkward and garbled sentence structures, and simply incorrect grammar. I do not have the time or inclination to identify or fix grammatical errors – I suggest the authors get help to identify and correct these errors, because they make it very difficult to evaluate the quality of the science being presented. Of greater concern is that the authors make many statements that seem to be entirely unsupported by any evidence or reference, and which appear unconnected to their main arguments. I have tried to identify these issues in the manuscript (see comments on specific parts, below), because it is imperative that these be corrected (or removed) for this work to be of publishable quality. In general, I recommend that the authors focus on their main argument (that the levels of variation they have identified have important implications for Proceratophrys taxonomy), and try to avoid getting side-tracked and making unsupported claims.

L47: “Understanding how evolutionary mechanisms operate on populations through morphological variability in individuals has been the main objective of the most different studies since the times of Darwin (Futuyma 1987; Huxley 1940).” This sentence does not make sense as written and needs to be modified. Did you mean “most studies”?
L91: “chromatic polymorphism provides these species with a varying amplitude which enables them to occupy, adapt and reproduce for generations in determined environments” It is not clear what is meant by “varying amplitude”. Please explain what you mean or remove this phrasing. Also, it is not clear what is meant by “determined environments”.
L96: “The adaptive importance of polymorphism for organisms, including P. cristiceps, lies in the improvement of the reproductive and survival capacities of individuals in response to stress caused by the environment or predators (Ridley 2004).” This is a very general statement about polymorphism that has only been validated in a limited number of species. Is there evidence that this is the case in P. cristiceps? If so, reference it, if not, you cannot use a general reference (a textbook) to claim evidence concerning what is going on in P. cristiceps (unless Ridley specifically discusses P. cristiceps, which I don’t think he does). In any case, if there is such evidence for P. cristiceps, you should cite the original references.
L99: “Regardless of the degree of changes, a limiting factor for survival is seen in the ability for morphophysiological modification, whether intraspecific or interspecific (Ricklefs 2008).” I do not understand what the authors are trying to say here – please rephrase and rewrite for clarity.
L101: “In this case, the survival and adaptation of organisms depends on the maintenance of a number of individuals in these populations in such a way that the evolutionary mechanisms can act on them, and thus give rise to what we call biodiversity” Again, it is not entirely clear what the authors are trying to get at here – rephrase.
L. 124: “Specimens Maturity” is awkward and grammatically incorrect – rephrase.
L. 134: A Chi Square test should only be used for comparisons of categorical data – it is not clear if that is what the authors did.
L. 139: “whose chromatic patterns were statistically functional with respect to the analysed frequencies.” I have no idea what this means – rewrite or rephrase.
L. 183: Change to “enabling us to rexamine…”
L. 192: “(not intercross)” – not clear what this means – explain or rewrite.
L. 193: “we decided to find presumably intercrossing sets aiming to mitigate methodological eventualities, or the “demes” so to speak” Again, I have no idea what the authors are trying to say here.
L. 245: “We used two software programmes with the goal of mitigating the possible effects caused by PM(g) and PB(g) in the BAM diagram (biotic, abiotic, and movements) of probabilities (errors of omission and commission) for species with restricted vagility”. I am not an expert on these methods, but I don’t think they are explained very well here. I recommend the authors take abit more space to explain these methods in more detail, making sure to explain any field-specific jargon.
L. 333: “Furthermore, we observed a proportional expression of the six phenotypes for each relative frequency of P. cristiceps (14:43:6:9:8:20), which was also maintained internally among the samples and localities”. It is not entirely clear to me what this means – I think further explanation is in order.
L. 401: “being exclusively distributed in the Caatinga and transition phyto physiognomies according to their area of habitat suitability.” Again, it is not clear to me what this means – please explain or rephrase.
L. 421: “When dealing with these probable varieties, it is necessary to identify categories which can be in themselves equivalent in an experimentally comparable way.” Not clear what this means?
L. 477: “random events involving skull conformation to food resources” Not clear what this means.
L. 485: “Furthermore, there are underlying factual (and experimental) requirements to explain the morphologic divergence and the alleged taxonomic diversity (Van Holstein & Foley 2020).” Not clear what is meant here – I think further explanation is in order.
L. 546: “but which less explain the proposed species,” Rephrase.
L. 566: Furthermore, they are not even tested under an experimental model of diversification dynamics (Ajmal Ali et al. 2014; Annibale et al. 2020; Schindel & Miller 2005; Van Holstein & Foley 2020)” I think this requires further detail and explanation.
L. 569: “and why not for the P. biggibosa, P. boei and P. appendiculata groups whose taxonomic history involves controversies and the use of equally variable phenetic traits? Where evidence of pre- or post-zygotic barriers or their biogeography continue to be elusive?” This sentence is garbled and it is not clear what the authors are trying to say. Reference is made to past controversies without explaining them.

Experimental design

The experimental design appears generally appropriate, but I would have liked more detail on how the Chi Square tests were carried out.

Validity of the findings

L. 428: “The variations observed in Proceratophrys cristiceps at the chromatic level indicated selective value in the studied species. This fact is reflected in the differential frequencies of the chromotypes that seem to signal some advantage of individual survival. The distribution of animals in the soil found herein suggests having a predominant role in the bias of chromotypic frequencies throughout post-larval development, i.e. it indicates a frequency-dependent selection (Bond 2007).” In my opinion, this paragraph is entirely speculative and should be removed or replaced with something much less assertive. As far as I can tell, the authors present no evidence that these different morphs are adaptive, other than a few pictures where some of the color morphs appear to blend into the background. This is not good evidence. Referencing a general review paper on predation and polymorphism by Bond also does not provide evidence. It is possible that these colors are an adaptive form of camouflage, but you need to carry out carefully controlled experiments to demonstrate that these color patterns are adaptive, not simply show a few pictures. The authors claim that the frequency distributions they observed somehow support the hypothesis of camouflage and frequency-dependence, but it is not at all clear how or why they came to this conclusion, and I did not see any valid test of distributions in the context of predictions of the camouflage hypothesis.

L. 450: “Therefore, we deduce that it may be a chromatic polymorphism with a strong genetic basis”. As far as I can tell, you do not have solid evidence for a genetic basis. You would need to do common garden experiments to demonstrate this, and those are not described in this paper.

Additional comments

Finally, the authors have put a number of the figures in the discussion. It is not clear why that is the case – typically figures are placed in the results. Personally, I find this approach confusing.

Reviewer 2 ·

Basic reporting

In general, this is a quite well-written article. I am not a native English speaker so I cannot fully assess the quality of the English but I think it's correct for the most part. However, some sentences are in my opinion unnecessarily wordy and thus difficult to read. There are also some typographic errors and some weirdly used words. Wherever I could, I suggested corrections (see the attached annotated PDF) but this is by no means an exhaustive check. Anyway, the text needs to be carefully checked again.

Experimental design

I believe that the methods used to tackle the problem of colour variability in Proceratophrys cristiceps are appropriate (although I am not familiar with some of the used methods such as ecological niche modelling). However, in a few cases, I would like some additional details which could help the reader to better understand what was done by the authors.

Lines 117-122. I think it would be beneficial to indicate - in the case of preserved specimens - how were they stored (e.g. frozen, stored in alcohol, formalin etc.). I do not see this information either in the main text or in the supplemental data. Also, if they were stored in alcohol, wouldn't this affect the results as the alcohol dehydrates the specimen and may change its colouration, especially after long storage?

Lines 124-142. I am not familiar with this group of amphibians so I have a question here. Is there a possibility to confuse P. cristiceps with another species? A cryptic congener maybe? If so, it would be good to indicate how the specimens were classified as P. cristiceps. An identification key or molecular analysis? If there are no other similar species, you can disregard this point.

Lines 265-274. Unfortunately, I do not understand how you achieved this. Please describe your procedure in more detail.

Validity of the findings

In my opinion, the main findings are well supported. However, please see the questions posed in the "Experimental design" section.

Additional comments

The manuscript authored by Kleber Silva Vieira, Erivanna Karlene Santos Oliveira, Washington Luiz Silva Vieira and Rômulo Romeu Nóbrega Alves aims to describe the variation in colouration and morphometrics in a South American amphibian Proceratophrys cristiceps. It is an important task because in most species the intraspecific variation is usually insufficiently known. This is quite unfortunate because, without good knowledge on intraspecific variation, it is difficult to correctly delimit the species and in consequence to use correct names which, of course, further hinders studies in many other aspects of biology. This contribution is thus needed and will be welcomed by other researchers studying the morphology of South American amphibians. To sum up, in my opinion the article will be acceptable after moderate revisions.

Some further comments:
Lines 1-3. I suggest changing the title to "Polymorphism in a Neotropical toad species: An ontogenetic, population and geographical approach to chromatic variation in Proceratophrys cristiceps (Müller, 1883) (Amphibia, Anura, Odontophrynidae)". I know that there are two brackets just next to each other and that does not look very aesthetically but I think that in this context it is justified. I also think that "Neotropical" should be capitalised (names of biogeographic realms are capitalised in English if I'm not mistaken).

Lines 173-174. "For example, morphometric variation in animal research can be identified as either sexual dimorphism or originated from ontogeny (allometry)" - That is quite confusing. The intersexual morphological differences can also result from allometry. I suggest removing the word "allometry", it is not needed in this sentence.

Line 178. Of course, the authors will make the final decision about this, but I suggest putting Figure S2 into the main article. What measurements were taken is very important information and in my opinion, it is better to give it in the main text rather than "hide" it in a supplement.

Line 470" "However, because they are anuran amphibians..." - This doesn't sound particularly well. Please, try rewriting. Maybe something more like this: "However, because there is no clear concordance with the Bergmann's rule in anurans..."

Line 477. "random events involving skull conformation to food resources" - I am not sure whether I understand this correctly but to me, the "skull conformation to food resources" is a result of the selective pressure related to the adaptation to certain diets and thus not a random event but is of presumably strictly adaptational nature. Please reword or indicate what events do you have in mind.

Lines 493. "Fig. 9 A" - to be honest, I don't think this figure is necessary in the main text. I would move it to the supplement but of course, the authors will make a final decision about that.

Lines 514-524. "Figure 10" - the description of this figure is too vague. There is a good chance that the reader will not understand this figure unless it is more clearly described (same as my comment about the lines 265-274).

Lines 526-535. "...our observations indicated that the nodules (including warts and tubercles) were extremely variable in terms of number, shape and distribution, either isolated or regionally, in the same individual or between specimens" - you do not give any specific data about that (or am I missing something?), nor cite previous works that support this claim.

Annotated reviews are not available for download in order to protect the identity of reviewers who chose to remain anonymous.

---

## Round 0.2 · Minor Revisions

Thank you for your careful revisions. There are just a few more changes needed to improve readability. I have marked them on the pdf attached. Please review them carefully, If you agree with them, accept the changes in Word and send the manuscript back to us. Thank you.

Reviewer 1 ·

Basic reporting

I am satisfied that the authors have made a sincere effort to improve the writing in this document. There are still some awkward constructions and grammatical errors here and there. I leave it to the editor to decide whether this demands further revision.

Experimental design

I am satisfied with the revisions.

Validity of the findings

I am satisfied with the revisions.

Additional comments

NA

---

## Round 0.3 · accepted · Accept

Thank you for making the changes.